# The Kappa Opioid Receptor Agonist 16-Bromo Salvinorin A Has Anti-Cocaine Effects without Significant Effects on Locomotion, Food Reward, Learning and Memory, or Anxiety and Depressive-like Behaviors

**DOI:** 10.3390/molecules28124848

**Published:** 2023-06-19

**Authors:** Ross van de Wetering, Amy Ewald, Susan Welsh, Lindsay Kornberger, Samuel E. Williamson, Bryan D. McElroy, Eduardo R. Butelman, Thomas E. Prisinzano, Bronwyn M. Kivell

**Affiliations:** 1School of Biological Sciences, Centre for Biodiscovery, Victoria University of Wellington, Wellington 6012, New Zealand; 2Department of Pharmaceutical Sciences, University of Kentucky, Lexington, KY 40506, USAprisinzano@uky.edu (T.E.P.); 3Department of Medicinal Chemistry, University of Kansas, Lawrence, KS 66045, USA; 4Laboratory on the Biology of Addictive Diseases, The Rockefeller University, New York, NY 10065, USA

**Keywords:** kappa opioid receptor, salvinorin A, cocaine, addiction, side effects

## Abstract

Kappa opioid receptor (KOR) agonists have preclinical antipsychostimulant effects; however, adverse side effects have limited their therapeutic development. In this preclinical study, conducted in Sprague Dawley rats, B6-SJL mice, and non-human primates (NHPs), we evaluated the G-protein-biased analogue of salvinorin A (SalA), 16-bromo salvinorin A (16-BrSalA), for its anticocaine effects, side effects, and activation of cellular signaling pathways. 16-BrSalA dose-dependently decreased the cocaine-primed reinstatement of drug-seeking behavior in a KOR-dependent manner. It also decreased cocaine-induced hyperactivity, but had no effect on responding for cocaine on a progressive ratio schedule. Compared to SalA, 16-BrSalA had an improved side effect profile, with no significant effects in the elevated plus maze, light–dark test, forced swim test, sucrose self-administration, or novel object recognition; however, it did exhibit conditioned aversive effects. 16-BrSalA increased dopamine transporter (DAT) activity in HEK-293 cells coexpressing DAT and KOR, as well as in rat nucleus accumbens and dorsal striatal tissue. 16-BrSalA also increased the early phase activation of extracellular-signal-regulated kinases 1 and 2, as well as p38 in a KOR-dependent manner. In NHPs, 16-BrSalA caused dose-dependent increases in the neuroendocrine biomarker prolactin, similar to other KOR agonists, at doses without robust sedative effects. These findings highlight that G-protein-biased structural analogues of SalA can have improved pharmacokinetic profiles and fewer side effects while maintaining their anticocaine effects.

## 1. Introduction

Substance use disorders (SUDs) exact an enormous medical, financial, and emotional toll on society, and there is an unmet need for new prevention and treatment strategies. Although it is now well established that SUDs are disorders of the brain, maintained, in part, by persistent changes in brain function, efficacious treatments for many SUDs have remained elusive [1,2]. Psychostimulant use disorders, in particular, still have no Food and Drug Administration (FDA)-approved treatments, despite their high prevalence.

Preclinical studies have demonstrated that the kappa opioid receptor (KOR) is a promising target for the management of psychostimulant use disorders. For example, the administration of KOR agonists reduced cocaine self-administration in rats, mice, and rhesus monkeys [3,4,5,6,7,8,9,10,11], reduced the cocaine- as well as amphetamine-primed reinstatement of drug seeking in rats [9,12,13,14], and blocked the development of locomotor sensitization as well as condition place preference to cocaine in rats [10,15,16,17,18]. These antipsychostimulant effects of KOR agonists have been attributed to the KOR-dependent modulation of central dopaminergic mechanisms, since microdialysis studies have shown that treatment with KOR agonists attenuated the cocaine- as well as amphetamine-induced synaptic overflow of dopamine (DA) in the nucleus accumbens (NAcc) [19,20,21,22,23]. The acute administration of KOR agonists also produced dose-dependent increases in DA re-uptake and the upregulation of cell surface DA transporters (DATs), suggesting that an interaction between a KOR and DAT might underlie these effects [20,24].

In contrast to agonists of the mu opioid receptor, selective KOR agonists have minimal abuse potential [25,26] and do not inhibit gastrointestinal transit [27] or cause respiratory depression [28,29]. These properties, in conjunction with their antinociceptive and immunomodulatory effects, also make them promising candidates for developing non-addictive pain medications [30]; however, negative side effects, such as aversion, anxiety, depression, anhedonia, and sedation, have been reported, which has limited their clinical development [11,31,32,33,34,35,36,37,38,39,40,41]. As such, in order to progress the development of KOR-targeted therapeutics, the minimization of these negative side effects and a better understanding of their underlying mechanisms are required [42,43,44,45].

One strategy for achieving this goal has been through the development of structurally novel KOR agonists with biased signaling profiles; that is, agonists that are capable of inducing unique receptor conformations that differentially activate various G-protein- and β-arrestin-dependent signaling pathways [46,47]. Accumulating evidence suggests that KOR agonists that show a bias towards the preferential activation of the G-protein pathway have fewer adverse side effects than agonists that show a bias for the β-arrestin pathway, while also retaining their therapeutic efficacy [48,49,50,51,52,53]. The above being the case, it should also be noted that it is difficult to consistently quantify and compare signaling bias due to the use of different cell lines, receptor species, and reference ligands, which have a significant impact on the assays [43,54]. Moreover, assays using modified proteins often do not accurately mimic the function of native proteins.

The neoclerodane diterpene salvinorin A (SalA) is a potent KOR agonist derived from the *Salvia divinorum* plant. SalA displays a moderate G-protein bias relative to U50,488 [55], but has also been shown to have balanced signaling properties when using different methods [52]. SalA has antipsychostimulant effects similar to those of prototypical KOR agonists such as U50,488 or U69,539 [14,56]; however, the rapid onset of action, short half-life (approximately 50 min in vivo), and pro-hallucinatory as well as dysphoric effects of SalA limit its direct clinical viability [57,58,59]. The value of SalA, nonetheless, has been as a structural scaffold for the synthesis of novel KOR agonists [60,61]. For example, SalA analogues derived from modifications at the C-2 position had greater metabolic stability than SalA, were more effective at attenuating cocaine-induced reinstatement as well as the expression of cocaine sensitization, and also had improved side effect profiles [55,62,63,64,65].

In the current study, we evaluate another SalA analogue, derived from bromination at the C-16 position (Figure 1) [60,66,67]. 16-bromo salvinorin A (16-BrSalA) has comparable affinity ([^3^H]diprenorphine binding: SalA K*_i_* = 2.5 ± 0.6 nM, 16-BrSalA K*_i_* = 2.9 ± 0.3 nM), efficacy (% [^35^S]GTP-γ-S binding relative to U50488: SalA EC_max_ = 105 ± 4, 16-BrSalA EC_max_ = 108 ± 3), and potency ([^35^S]GTP-γ-S binding: SalA EC_50_ = 2.1 ± 0.6 nM, 16-BrSalA EC_50_ = 2.4 ± 0.2 nM; inhibition of cAMP [HitHunter]: SalA EC_50_ = 0.03 ± 0.004 nM, 16-BrSalA EC_50_ = 0.04 ± 0.01 nM) to SalA, indicating that the C-16 position does not interact with the KOR [60,67]. 16-BrSalA has a G-protein signaling bias in vitro and showed promising results in our previous studies, decreasing the cocaine-induced reinstatement of drug-seeking behavior without impacting spontaneous locomotor activity or anxiety in vivo [53,60]. We expand on these findings here, determining the duration of action of 16-BrSalA and further investigating the anticocaine effects of 16-BrSalA in vivo. We also evaluate 16-BrSalA for a range of side effects, using preclinical models for anxiety and depressive-like behavior, learning and memory, anhedonia, and aversion. Furthermore, in order to investigate some of the mechanisms underlying the effects of 16-BrSalA, we examine the effect of 16-BrSalA on DAT function, as well as second messenger signaling via mitogen-activated protein kinases (MAPKs), extracellular-signal-regulated kinases 1 and 2 (ERK1/2), and p38. Lastly, we also compare the effects of 16-BrSalA and SalA in non-human primates with high KOR homology to humans [68] on a translational neuroendocrine biomarker (prolactin levels).

## 2. Results and Discussion

### 2.1. 16-BrSalA Has a Longer Duration of Action Compared to SalA In Vivo

KOR agonists, including SalA and its analogues, have well-established antinociceptive effects [43,53]. As such, a warm water tail withdrawal assay was used to evaluate the in vivo duration of action of 16-BrSalA compared with SalA (Figure 2). A two-way ANOVA on the time course data revealed a significant treatment × time interaction, *F*(18, 234) = 2.21, *p* = 0.0039. Follow-up comparisons showed that treatment with SalA had significant antinociceptive effects primarily within the first 15 min, whereas 16-BrSalA had significant effects for over 90 min (*p* < 0.05; Figure 2a). A one-way ANOVA on the area under the curve (AUC) data similarly revealed a significant effect of treatment, *F*(2, 26) = 6.36, *p* = 0.0056, with 16-BrSalA, but not SalA, having a significant overall antinociceptive effect during these 150 min compared to vehicle-treated controls (*p* < 0.05; Figure 2b). These results are consistent with our previous, independent observations that used 16-BrSalA as well as another C16-based analogue, 16-ethynyl SalA [53]. These findings demonstrate that C-16 modifications to SalA can improve pharmacokinetic profiles, which are one of the primary limitations of SalA and its analogues as therapeutic agents. We have previously shown that SalA analogues derived from modifications at the C2 position similarly have a longer duration of action in antinociceptive assays, with improved metabolic stability in vivo [65]. This has been attributed to the lack of a hydrolysable ester [69]. How C16 alterations of SalA impact pharmacokinetic profiles have yet to be thoroughly investigated, however.

### 2.2. 16-BrSalA Attenuates Cocaine-Primed Reinstatement and Hyperactivity, but Not Progressive Ratio Responding

The anticocaine effects of 16-BrSalA are shown in Figure 3. As has been previously shown [60], 16-BrSalA produced a dose-dependent attenuation of cocaine + cue-induced reinstatement of drug-seeking behavior (Figure 3a). A one-way repeated measures ANOVA revealed a significant effect of treatment, *F*(3, 15) = 8.35, *p* = 0.0017, with SalA and both doses of 16-BrSalA significantly attenuating total active lever responses compared to vehicle-treated controls (*p* < 0.05). Of note, while a low dose of 16-BrSalA (0.3 mg/kg) produced a significant attenuation of drug-seeking behavior, a larger dose (1 mg/kg) was required to produce effects equivalent to SalA (0.3 mg/kg), despite both compounds having a similar affinity, potency, and efficacy for the KOR in vitro [60,67]. This difference might be due to the greater potency of SalA in its ability to modulate DAT function compared to 16-BrSalA (discussed in the next section).

To determine if these anticocaine effects of 16-BrSalA were KOR-dependent, the effect of cotreatment with the KOR antagonist norbinaltorphimine (nor-BNI) was also determined in a subset of animals (Figure 3b). A two-way fully repeated measures ANOVA on the time course data revealed a significant treatment × time interaction, *F*(46, 138) = 3.92, *p* < 0.0001. Follow-up comparisons showed that 16-BrSalA treatment produced a significant attenuation of responding from 115 to 120 min (*p* < 0.05), which was reversed by nor-BNI treatment (*p* < 0.05), indicating KOR dependency. A one-way repeated measures ANOVA on the summed response data revealed a significant effect of treatment, *F*(2, 4) = 20.71, *p* = 0.0078, with 16-BrSalA also significantly attenuating total responding in a KOR-dependent manner (*p* < 0.05).

The effect of 16-BrSalA on cocaine-produced locomotor hyperactivity is shown in Figure 3c. The cocaine prime was the same dose (20 mg/kg) as that used during the reinstatement tests. A two-way ANOVA on the time course data revealed a significant treatment × time interaction, *F*(34, 187) = 2.50, *p* < 0.0001, with significant decreases in ambulatory counts as a function of 16-BrSalA treatment at 15 and 20 min (*p* < 0.05), which was prevented by nor-BNI pretreatment. The total cocaine-produced ambulatory counts were also significantly decreased via 16-brSal treatment in a KOR-dependent manner (*p* < 0.05). These results are consistent with those produced by other KOR agonists, including SalA and analogues [55]. Of note, we have previously shown that 16-BrSalA did not decrease spontaneous locomotor activity [60]; therefore, these results are unlikely to be due to a generalized decrease in motor activity, but rather 16-BrSalA counteracting the locomotor-activating effects of cocaine.

These results suggest that, as with other KOR agonists, 16-BrSalA was effective at attenuating the cocaine-induced reinstatement of drug-seeking behavior as well as cocaine-induced locomotor activity. To further expand on these findings, we also determined for the first time if KOR activation via 16-BrSalA or other KOR agonists was similarly effective at attenuating the reinforcing effects of cocaine on a progressive ratio (PR) schedule, where the response requirement increases with each successive infusion (Figure 3d). A one-way repeated measures ANOVA revealed a significant effect of treatment, *F*(6, 78) = 4.88, *p* = 0.0003, which was driven by a significant attenuation of PR responding produced by U69,593 (*p* < 0.05). No significant effects from other treatments were found. This might be due to the longer session duration used during the PR tests (up to 5 h) and the relatively short duration of action of KOR agonists. SalA has been previously shown to significantly decrease breakpoints for sucrose self-administration, but this was during 2 h sessions [70]. In rhesus monkeys, SalA and nalfurafine both decreased responding for oxycodone on a PR schedule; however, in this study SalA/nalfurafine was mixed with oxycodone infusions, rather than administered separately at the start of the session [71].

### 2.3. 16-BrSalA Increases DAT Activity in a KOR- and ERK-Dependant Manner

The activation of central dopaminergic mechanisms is crucial for the acute reinforcing as well as locomotor-activating effects of cocaine and other psychostimulants [72,73]. Microdialysis studies have shown that KOR agonists can attenuate cocaine-induced increases in synaptic DA concentrations in the NAcc, and, as such, this has been suggested to be a mechanism underlying the antipsychostimulant effects of KOR agonists [19,20,21,22,23]. We have previously shown that SalA and some analogues can increase DAT activity [24,62]. Here, we determined whether 16-BrSalA would have the same effect (Figure 4).

DA uptake kinetics across 0–4 µM concentrations was measured in both NAcc (Figure 4a) and dorsal striatal (dSTR; Figure 4b) tissue suspensions via the use of rotating disk electrode voltammetry (RDEV). DAT activity was also determined in individual HEK-293 cells coexpressing DAT and KOR (Figure 4c) via the use of time-resolved imaging of the fluorescent monoamine transporter substrate 4-[4-(dimethylamino)styryl]-*N*-methylpyridinium (ASP^+^). The calculated V_max_ and K_m_ values from these uptake studies are shown in Table 1. 16-BrSalA significantly increased V_max_ in both NAcc (*t*(16) 2.41, *p* = 0.0282) and dSTR (*t*(16) = 2.25, *p* = 0.0386) tissue suspensions, but had no effect on K_m_. In cells, 16-br SalA significantly increased both V_max_ (*t*(498) 4.15, *p* < 0.0001) and K_m_ (*t*(498) 4.06, *p* < 0.0001). These findings indicate that, as with SalA, 16-BrSalA increased DAT activity, which would be expected to effectively decrease synaptic DA concentrations and attenuate the effects of cocaine.

To confirm the KOR-dependent effect of 16-BrSalA on DAT activity, the effect of pretreatment with nor-BNI on the uptake of fixed concentrations of DA was also determined. Separate one-way ANOVAs revealed significant effects of treatment on both NAcc (*F*(2, 24) = 8.62, *p* = 0.0015) (Figure 4d) and dSTR (*F*(2, 21) = 6.05, *p* = 0.0084) (Figure 4e) tissue suspensions. Follow-up tests indicated that 16-BrSalA significantly increased uptake in both assays (*p* < 0.05), which was blocked by nor-BNI cotreatment (*p* < 0.05), indicating KOR dependency.

In cells, we compared the effect of SalA and two doses of 16-BrSalA on the uptake of fixed concentrations of ASP^+^ and determined the effect of pretreatment with nor-BNI or the ERK1/2 inhibitor, U0126 (Figure 4f). A one-way ANOVA revealed a significant effect of KOR agonist treatment, *F*(3, 224) = 12.83, *p* < 0.0001, with both SalA and the higher dose 16-BrSalA significantly increasing ASP^+^ uptake (*p* < 0.05). Separate two-way ANOVAs revealed a significant interaction between 16-BrSalA treatment and U0126 treatment, *F*(1, 204) = 7.28, *p* = 0.0076, and a trending interaction between 16-BrSalA treatment and nor-BNI treatment, *F*(1, 195) = 3.15, *p* = 0.078. Follow-up comparisons indicated that 16-BrSalA-induced increases in ASP^+^ uptake were both KOR- and ERK1/2-mediated (*p* < 0.05). ERK1/2 regulates the transport capacity and intracellular trafficking of the DAT in the striatum [74]. We have previously shown that SalA-induced increases in DAT activity were mediated by ERK1/2 [24,62]. The data we collected using 16-BrSalA are consistent with this and support the idea that the anticocaine effects of KOR agonists are, in part, mediated through ERK-dependent DAT modulation. These data also show that SalA is more potent than 16-BrSalA at modulating DAT activity, which might explain why a larger dose of 16-BrSalA was required to produce anticocaine effects comparable to SalA (Figure 3a).

### 2.4. 16-BrSalA Has an Improved Side Effect Profile Compared to SalA

In order to develop clinically viable KOR-targeted treatments, it is vital to identify novel compounds that have reduced side effects compared to U50,488 or SalA. 16-BrSalA shows a greater G-protein signaling bias compared to SalA [53,55] as well as a more stable pharmacokinetic profile (Figure 2), and was, therefore, predicted to produce fewer adverse side effects. This was tested in the current study (Figure 5). Of note, the dose of 16-BrSalA evaluated in these side effect assays was the dose (1.0 mg/kg) that was required to produce significant anticocaine effects equivalent to SalA (0.3 mg/kg; Figure 3).

To evaluate the potential anxiogenic effects of 16-BrSalA, we used two preclinical models: the elevated plus maze and the light–dark test. A one-way ANOVA on the elevated plus maze data revealed a significant effect of treatment, *F*(3, 75) = 3.46, *p* = 0.0205, with both doses of SalA, but not 16-BrSalA, significantly decreasing the time spent on the open arms of the maze (*p* < 0.05; Figure 5a). Similar results were found in the light–dark test (Figure 5b): a one-way ANOVA produced a significant effect of treatment, *F*(3, 55) = 5.96, *p* = 0.0014, but with only the higher dose of SalA (1.0 mg/kg) having a significant effect (*p* < 0.05). These results indicate that, in contrast to the parent compound, 16-BrSalA does not have significant anxiogenic effects at these doses, which is consistent with our previous observations in mice using the elevated zero maze and the marble burying task [53]. Of note, neither agonist impacted the total distance travelled during the light–dark test (*p* > 0.05). Moreover, we have previously shown that 16-BrSalA (1 mg/kg) or SalA (0.3 mg/kg) had no effect on spontaneous locomotor activity in rats [14,60], suggesting that 16-BrSalA does not induce sedation and that sedation was not a confound in the current experiments.

Depressive-like behavior, anhedonia, and reduced non-drug reward behavior are related side effects associated with KOR activation. These effects have been primarily attributed to KOR-mediated decreases in central DAergic activity [6,70,75]. Given that this has been the proposed mechanism through which KOR agonists attenuate the effects of cocaine and other psychostimulants, finding a therapeutic window where KOR agonists can produce antipsychostimulant effects without these side effects is critical. In the current study we examined the effect of 16-BrSalA on the forced swim test, a measure of learned helplessness and depressive-like behavior, and on reinforcement maintained by a sucrose reward. In contrast to what has been previously shown with SalA or other SalA analogues [14,55,64], there was no effect of 16-BrSalA treatment on mobility or immobility time in the forced swim test (*p* > 0.05; Figure 5c). Similarly, 16-BrSalA had no effect on sucrose reinforcement (Figure 5d); a one-way repeated measures ANOVA revealed a significant effect of treatment, *F*(5, 30) = 60.89, *p* < 0.0001, but this was driven by a large attenuation of responding produced by U50,488 and a small increase produced by U69,593 (*p* < 0.05). This effect of U50,488 treatment is consistent with previous observations [14,76] and could be driven by the sedative effects of U50,488 [77] or differences in DA-independent mechanisms related to satiety and food consumption [78,79,80].

KOR activation has been known to disrupt processes related to learning and memory, with stress-induced impairments playing a particularly important role [81,82,83,84,85]. Here, we show that 16-BrSalA had no effect on learning and memory, as assessed by the novel object recognition task (*p* > 0.05; Figure 5e). This might be due to a lack of stress induced by 16-BrSalA, given that there was no effect of 16-BrSalA in our anxiety or forced swim tests; however, the further evaluation of both behavioral and biological markers of stress (i.e., corticotrophin-releasing factor) would be needed to confirm this. In our previous report, neither SalA nor mesyl salvinorin B affected novel object recognition [55], although significant effects of SalA on other measures of learning and memory, such as the eight-arm radial maze and passive avoidance, have been reported [86]. U50,488, on the other hand, has been previously shown to cause deficits in the novel object recognition task [84] and showed a trending effect in our study (*p* = 0.0860), which could be stress-induced [81].

Lastly, we determined the aversive properties of 16-BrSalA in a conditioned place aversion (CPA) paradigm (Figure 5f). As with other SalA analogues previously tested [65], 16-BrSalA treatment had significant conditioned aversive effects in our tests, as did SalA and U50,488 (*p* < 0.05). There is evidence to suggest that KOR-induced aversion requires β-arrestin and, more specifically, p38 signaling [87,88]. White and colleagues, however, found that conditioned place aversion to U50,488, U69,593, and SalA was still produced in β-arrestin knockout mice [52]. On the other hand, studies have shown that KOR-mediated aversion also involves interactions with both dopaminergic and serotoninergic mechanisms [89,90]. These studies suggest that the mechanisms underlying KOR-induced aversion are multifaceted; more research on the development of non-aversive KOR agonists is needed.

### 2.5. Effect of 16-BrSalA on ERK1/2 and p38

KOR agonism activates many signal transduction cascades, namely MAPK pathways, which include ERK1/2 and p38. As shown previously, as well as in the current study, the KOR-induced modulation of the DAT is mediated by ERK1/2, although no differences in the KOR modulation of ERK1/2 have been identified. KOR-induced ERK1/2 activation has both β-arrestin- and G-protein-dependent mechanisms, with increases in early phase ERK1/2 activation (5–15 min) being G-protein-dependent and late-phase ERK1/2 activation (~2 h) being β-arrestin-dependent [91,92]. Here, we quantified phosphorylated ERK1/2 levels in rat NAcc (Figure 6a), dSTR (Figure 6b), and prefrontal cortex (PFC; Figure 6c) tissue, as well as in HEK-293 cells coexpressing the DAT and KOR (Figure 6d), at different time points following 16-BrSalA treatment. The aim was to determine if, firstly, 16-BrSalA increased ERK1/2 activation, and, secondly, if the time course of activation reflects that of a G-protein-biased agonist. Peak increases in ERK1/2 phosphorylation were observed 10–15 min following 16-BrSalA treatment, though these increases were only significant in cells (*p* < 0.05). In order to determine if this effect was KOR- and ERK1/2-dependent, the effect of nor-BNI or U0126 cotreatment on 16-BrSalA-induced changes in phosphorylated ERK1/2 levels after a 10-minute incubation was also determined in cells (Figure 6e). Relative to vehicle treatment, 16-BrSalA significantly increased ERK1/2 phosphorylation, which was blocked by nor-BNI, while U0126 abolished all ERK1/2 activity (*p* < 0.05). The full Western blot membranes from Figure 6 are shown in Appendix A.

We also examined the effect of 16-BrSalA on p38 phosphorylation over time in rat NAcc (Figure 6f), dSTR (Figure 6g), and PFC tissue (Figure 6h). Because p38 phosphorylation is largely dependent on β-arrestin recruitment [87], G-protein-biased agonists, such as 16-BrSalA, would be expected to produce minimal effects. We found small increases in all brain regions, which peaked at 15 min, but this was only statistically significant in the NAcc, *t*(4) 4.23, *p* < 0.0134. The KOR-induced activation of p38 has been suggested to contribute to some of the undesirable side effects of KOR agonists [44,87,93,94]. The small but significant increase in p38 activity in the NAcc following 16-brSalA treatment might therefore contribute to the conditioned aversive response observed in the current study.

### 2.6. Effect of 16-BrSalA on Prolactin as a Neuroendocrine Biomarker in Non-Human Primates

Lastly, in order to provide translational data on 16-BrSalA, we evaluated the effect of 16-BrSalA on prolactin levels in blood samples collected from male rhesus monkeys (*n* = 3). Peripheral blood levels of prolactin can be used as a translationally valid, quantitative neuroendocrine biomarker of KOR-mediated effects, including the potency and apparent efficacy of ligands [95,96,97]. Figure 7a shows the cumulative dose–effect curves produced by 16-BrSalA, SalA, the partial KOR agonist nalorphine, and the vehicle on prolactin levels. Figure 7b shows the duration of effect of 16-BrSalA and Figure 7c shows the sedation score. Compared with Sal A, 16-BrSal A was approximately 0.5 log units less potent at inducing the release of prolactin, with similar maximal effects, larger than those of the KOR partial agonist nalorphine. Importantly, at doses that produced robust neuroendocrine effects, 16-BrSalA only produced slight sedation. The difference in potency between 16-BrSalA and SalA is comparable with the effects observed in drug-seeking models in rats (Figure 3a), anxiety models in rats (Figure 5a,b) as well as mice [53], motor incoordination tests in mice [53], and DAT modulation in vitro (Figure 4f). Together, these would suggest that the differences in the effectiveness of 16-BrSalA and SalA to produce several relevant effects may be similar across species.

## 3. Materials and Methods

### 3.1. Subjects

Adult male B6-SJL mice (23–26 g) were used in the tail withdrawal assay. Mice were obtained from the Malaghan Institute of Medical Research (Wellington, New Zealand) and housed within the School of Biological Sciences animal facility, Victoria University of Wellington (Wellington, New Zealand). Adult male Sprague Dawley rats (300–400 g) were used in all other rodent experiments. Rats were bred and housed within the School of Biological Sciences or the School of Psychology animal facilities (Wellington, New Zealand). Food and water were available ad libitum for all of the rodents, except for the rats in the sucrose self-administration experiment, which were maintained at 85% of their feeding weight via food restriction. Mice were housed in groups of 4–5, while rats were housed either individually (cocaine self-administration) or 2–3/cage (all other experiments). The animal facilities were temperature (19–21 °C)- as well as humidity (55%)-controlled and on a 12 h light/dark cycle (lights on at 07:00), with testing conducted during the light cycle. All experimental protocols involving rodents were approved by the Victoria University of Wellington Animal Ethics Committee, New Zealand.

Adult, gonadally intact, and captive-bred male rhesus monkeys (*Macaca mulatta*, weight range: 9–12 kg) were used for the prolactin assay. They were singly housed in a stable colony room maintained at 20–22 °C with controlled humidity and lights on from 07:00 to 19:00. Monkeys had visual, auditory, and olfactory access to other conspecifics, and an environmental enrichment plan was in place. They were fed appropriate amounts of primate chow (PMI Feeds, Richmond, VA, USA), supplemented by treats. Water was available ad libitum. Consecutive experiments in the same subject were typically separated by at least 72 h, with all experiments carried out between 10:00 and 14:00 h. All experiments involving non-human primates were approved by the Rockefeller University Animal Care and Use Committee, in accordance with the Guide for the Care and Use of Animals (National Academy Press; Washington, DC, USA).

### 3.2. Drugs and Treatment

SalA was isolated and purified from *Salvia divinorum* leaves as previously described [68]. 16-BrSalA was synthesized as previously described [60]. U50,488, U69,593, U0126, and DA were purchased from Sigma Aldrich (St. Louis, MO, USA). ASP^+^ was purchased from Tocris Biosciences (supplied by Pharmaco NZ Ltd., Auckland, New Zealand). KOR agonists were suspended in 75% DMSO for reinstatement, locomotor activity, and forced swim tests, as well as for ex vivo tissue collection. A vehicle with a 4:1:5 ratio of propylene glycol:dimethyl sulfoxide (DMSO):phosphate-buffered saline (PBS) was used for the tail withdrawal procedure, and a 2:1:7 ratio of DMSO:Tween-80:water was used for all of the other in vivo tests in rodents. A vehicle containing a 1:1:8 ratio of ethanol:Tween-80:water was used for the prolactin assay. For rodent tests, all of the agonists were administered at a volume of 1 mL/kg intraperitoneally (i.p.), except for U69,593, which was administered subcutaneously (s.c.). Agonists were administered either 5 min (SalA), 10 min (16-BrSalA, U50,488), or 15 min (U69,593) prior to testing based on previous work [14,56] and from the results of the duration of action experiment in the current study. For the prolactin assay, agonists were administered intravenously (i.v.) at a volume of 0.05–0.1 mL/kg. The KOR antagonist nor-BNI was dissolved in 0.9% NaCl and administered subcutaneously 24 h prior to testing at a volume of 1 mL/kg. Cocaine-HCl (BDG synthesis; Wellington, New Zealand) was dissolved in 0.9% NaCl containing sodium heparin (3 U/mL) for intravenous (i.v.) infusions or in 0.9% NaCl for i.p. priming injections. For ex vivo and in vitro experiments, 16-brSal A, nor-BNI, and U0126 were dissolved in 100% DMSO and diluted in a KREBS buffer (130 mM NaCl, 1.3 mM KCl, 2.2 mM CaCl_2_, 1.2 mM MgSO_4.6_H_20_, 1.2 mM KH_2_PO_4_, 10 mM HEPES, and 10 mM D-glucose, pH 7.4) to reach their working drug concentrations, resulting in vehicle DMSO concentrations of 0.001% (RDEV) or 0.02% (all other experiments) for 16-brSal A, and 0.01% and 0.2% for nor-BNI and U0126, respectively.

### 3.3. Warm Water Tail Withdrawal Assay

The warm water tail withdrawal assay was carried out as previously described [62]. Briefly, mice were restrained in a transparent plexiglass tube (internal diameter of 24 mm) and allowed to acclimatize for 15 min. The distal 1/3 portion of the tail was then immersed in a water bath (50 ± 0.5 °C), and the latency to withdrawal was measured (maximum time allowed = 10 s). Following baseline measurements (average of 3), the effect of vehicle (*n* = 10), SalA (1.0 mg/kg, *n* = 8), and 16-BrSalA (1.0 mg/kg, *n* = 11) treatment was determined. Measurements were taken at 1, 5, 10, 15, 30, 45, 60, 90, 120, and 150 min postinjection. The maximum possible effect (MPE) of antinociception was calculated as follows: MPE (%) = 100 × (test latency − control latency)/(10 − mean control latency).

### 3.4. Surgery and Cocaine Self-Administration

Indwelling intravenous catheters were surgically implanted in the external jugular vein as previously described [14]. Rats were trained to self-administer in standard operant chambers equipped with two levers (Med Associates, Fairfax, VT, USA; model ENV-001) and a mechanical syringe pump (Med Associates, USA; model PHM100A). The depression of the active lever resulted in a 0.1 mL i.v. infusion of cocaine (0.5 mg/kg/infusion) over 12 s and the illumination of a light located above the lever. The depression of the inactive lever resulted in no planned consequence. Drug delivery and recording the number of active/inactive lever responses made were controlled by Med PC software (v4.2, Med Associates, Fairfax, VT, USA).

All self-administration sessions were conducted during 2 h sessions, 6 days per week. Rats were initially trained on a fixed-ratio (FR) 1 schedule of reinforcement until responding had stabilized at ≥20 infusions per session with an active:inactive ratio of ≥2:1 for 3 consecutive sessions. Thereafter, the schedule of reinforcement was increased to FR2 until this criterion had been reached once more before the schedule of reinforcement was increased again to FR5. Rats were run for 10 days at FR5 to establish stable baseline responding before beginning the reinstatement or PR experiments.

As has been previously reported [60], the reinstatement experiment was run in three repeating phases. In phase 1 of reinstatement (baseline), rats (*n* = 6) self-administered cocaine on an FR5 schedule for at least 2 days, until their responses were within 20% of their prereinstatement baseline. In phase 2 (extinction) the light cue was removed and cocaine was substituted for the vehicle until responses had dropped to <20 within a single session (3–4 days). In phase 3 (reinstatement), rats received either the vehicle, SalA (0.3 mg/kg), or 16-BrSalA (0.3 or 1.0 mg/kg) prior to a priming injection of cocaine (20 mg/kg). The light stimulus that was previously paired with cocaine infusions was also reintroduced, and the number of responses made was recorded. Treatment was administered in a within-subjects Latin square design. The effect of nor-BNI (10 mg/kg) treatment combined with 16-BrSalA (1.0 mg/kg) was also determined in three of these animals, but this was carried out as the final test due to the long-lasting effects of nor-BNI. No significant differences in baseline or extinction responding were observed in-between reinstatement treatments (*p* > 0.05). The mean inactive lever responding ranged from 1.5 to 2.3 across treatments, and similarly did not significantly differ (*p* > 0.05).

For the PR experiment, rats (*n* = 14) were administered either the vehicle, U50,488 (10.0 mg/kg), U69,593 (1.0 mg/kg), SalA (0.3, 1.0 mg/kg), or 16-BrSalA (1.0, 2.0 mg/kg) prior to self-administering cocaine, where the response requirement for reinforcement increased according to the following equation:
$$5e^{\left(0.2\;\times \;\mathrm{infusion}\#\right)}-5\;\left(\mathrm{rounded}\;\mathrm{to}\;\mathrm{nearest}\;\mathrm{interger}\right)$$
which results in the following ratios: 1, 2, 4, 6, 9, 12, 15, 20, 25, 32, 40, 50, 62, 77, etc. [98]. The session continued until 60 min had elapsed without drug infusion. Treatment was administered following a within-subjects Latin square design. Prior to each treatment session, rats were subjected to at least 3 days of PR responding to establish a stable breakpoint (±2 infusions for 3 consecutive days).

### 3.5. Cocaine-Induced Locomotor Activity

Locomotor activity was measured in clear plexiglass chambers (42 × 42 × 30 cm; Med Associates, USA; model ENV-515) set in sound-attenuating boxes. Each chamber contained a lattice of 32 infrared beams, 1.7 cm above the floor of the chamber. The sequential interruption of 3 beams was recorded as one ambulatory count. Counts were recorded in 5 min bins via activity-monitoring software (Med Associates, USA). All locomotor experiments were conducted in the dark and in the presence of white noise. The rats were placed into locomotor activity chambers for 30 min prior to receiving either the vehicle (*n* = 6) or 16-BrSalA (1.0 mg/kg, *n* = 6). Another group of rats were pretreated with nor-BNI (10 mg/kg) before receiving 16-BrSalA (1.0 mg/kg; *n* = 2). Thereafter, all of the rats received an injection of cocaine (20 mg/kg) and were returned to the locomotor activity chambers for an additional 60 min. Ambulatory counts were recorded during the entire 90 min.

### 3.6. Light–Dark Test

A large white chamber (30 × 30 × 34 cm) connected to a smaller black chamber (15 × 30 × 34 cm) via a small grey corridor (8 × 10 × 34 cm) was used to carry out the light–dark tests. Testing was conducted in a dark room with three LED lamps: one directed at the light box, one at the corridor, and one at the ceiling. The intensity of the light in each chamber was as follows: 100 lux in the light chamber, 10 lux in the dark chamber, and 70 lux in the corridor. The rats were treated with either the vehicle (*n* = 24), SalA (0.3 mg/kg, *n* = 13; 1.0 mg/kg, *n* = 13), or 16-BrSalA (1.0 mg/kg, *n* = 9), and placed in the black chamber. The time spent in the light box as well as the total distance travelled were then determined during 15 min of free access. Activity was measured via the use of SMART 3.0 software (Panlab, Harvard Apparatus, Holliston, MA, USA).

### 3.7. Elevated plus Maze

The EPM maze consisted of four arms (50 cm long × 10 cm wide) elevated 55 cm above the ground. Two of the arms had a small parapet measuring 2.5 cm in height (open arms), while the other two arms were enclosed by 40 cm-high black walls (closed arms). The rats were habituated to the conditions in the testing room for 60 min and then treated with either the vehicle (*n* = 29), SalA (0.3 mg/kg, *n* = 16; 1.0 mg/kg, *n* = 19), or 16-BrSalA (1.0 mg/kg, *n* = 15), before being placed in the center of the apparatus facing an open arm. All subsequent activity was recorded for 5 min via the use of a Sony HDR-SR5E digital camera recorder, and the time spent in each arm was calculated by an observer blinded to the experimental treatment. Open arm time was calculated only when the rats had all four paws in the open arm.

### 3.8. Forced Swim Test

The forced swim test was conducted according to the methods described in [99], using a cylindrical chamber measuring 44 cm in height and 20 cm in diameter, which was filled with water (25 ± 1 °C) to a height of 35 cm. Testing was conducted over 2 days. On both days, rats were habituated to the testing room for 60 min. On day 1, drug-naïve rats were habituated to forced swimming conditions for 15 min. On day 2, rats received either vehicle (*n* = 8) or 16-BrSalA (1.0 mg/kg, *n* = 9) treatment before a 5 min test session. Sessions were recorded via the use of a SONY HDR-SR5E digital camera recorder and analyzed for mobility as well as immobility time, which was measured in 5 sec intervals by an observer blinded to experimental treatments.

### 3.9. Sucrose Self-Administration

The rats maintained at 85% of their free-feeding weight were trained to self-administer sucrose pellets (Dustless Precision Pellet, 45 mg sucrose; Able Scientific, Perth, Australia) in standard operant chambers (Med Associates, USA, ENV-011) with two levers: one lever was connected to a sucrose pellet dispenser and a light (active lever), while the second lever had no programmed function (inactive lever). Sucrose self-administration sessions were conducted during 45 min sessions, 6 days per week. Rats (*n* = 7) were initially trained on an FR1 schedule of reinforcement until they had self-administered ≥20 pellets during a single session with an active:inactive ratio of ≥2:1. Thereafter, rats progressed to an FR5 schedule for at least 5 days to establish stable baseline response rates. The effect of the vehicle, U50,488 (10 mg/kg), U69,593 (0.3 mg/kg), SalA (0.3 mg/kg), or 16-BrSalA (1.0 mg/kg) on sucrose self-administration was then determined in a within-subjects Latin square design. Sucrose delivery and the recording of the number of lever responses made were controlled via Med PC software (Med Associates, USA).

### 3.10. Novel Object Recognition

The novel object recognition test was carried out based on previously published methods [85]. On days 1–3 (habituation), rats (*n* = 21) were habituated to the testing chamber (45 × 45 × 35 cm, open field) for 30 min/day. On day 4 (familiarization), rats were familiarized with two identical objects, which were introduced on either side of the chamber (3 times for 6 min, with an inter-trial interval of 10 min). On day 5 (test), one of the familiar objects was replaced with a novel object, and the time spent interacting with both objects was recording via the use of SMART 3.0 software (Panlab, Harvard Apparatus, Holliston, MA, USA). The recognition index was calculated as follows:
$$\left(\frac{N}{F+N}\right)\times 100$$
where *N* is the time (in seconds) spent with the novel object and *F* is the time spent with the familiar object. Rats that only interacted with a single object were excluded. The effect of treatment with either the vehicle, U50,488 (10 mg/kg), SalA (0.3, 1.0 mg/kg), or 16-BrSalA (1.0 mg/kg) was determined via the use of a within-subjects Latin square design.

### 3.11. Conditioned Place Aversion

CPA was conducted via the use of a biased procedure based on previously published methods [100]. A 3-chamber apparatus was used (PanLab, Harvard Apparatus, USA), which had two large chambers (30 × 30 × 34 cm) connected by a small corridor (8 × 10 × 34 cm) with removable sliding doors. One of the large chambers had a textured black floor with black dotted patterns on its walls (black chamber), and the other had a smooth white floor with black striped patterns on its walls (white chamber). The corridor was a neutral zone with grey walls as well as floor, and illuminated at an intensity of 70 lux. The average light intensity in both conditioning chambers was 20 lux. Experiments were conducted in the presence of white noise.

The CPA procedure was conducted over 9 days. On day 0, rats were habituated to the CPA apparatus for 15 min. On day 1 (preconditioning), rats were given free access to all of the chambers for 15 min. Animals that showed >80% preference for a particular chamber or >40% preference for the corridor were excluded. On days 2–7 (conditioning), rats were treated with either the vehicle (*n* = 9), U50,488 (10 mg/kg, *n* = 11), SalA (0.3 mg/kg, *n* = 9), or 16-BrSalA (1.0 mg/kg, *n* = 9) and confined in their preferred chamber for 45 min. On alternating days, in a counterbalanced order, rats received the vehicle and were confined to their less preferred chamber for 45 min. On day 8 (postconditioning), rats were placed in the corridor and given free access to the apparatus for 15 min. The time spent in each chamber was recorded during the pre- and postconditioning sessions via the use of SMART 3.0 software (Panlab, Harvard Apparatus, Holliston, MA, USA).

### 3.12. Prolactin Assay

Following extensive habituation and chair training, an indwelling catheter (27 gauge, Surflo; Terumo, Tokyo, Japan) was placed percutaneously in a saphenous vein and attached to a Luer multisample injection plug (the catheter was removed at the end of each experiment). The catheter and plug were flushed with 0.3 mL of heparinized sterile saline (50 U/mL) prior to use, as well as after each injection or sampling. Approximately 15 min following catheter placement, two baseline blood samples were obtained (approximately 2 mL each). These baseline samples were collected 5 min apart from each other (approximately −10 and −5 min, relative to the onset of dosing). These blood samples were placed in a plain vacutainer and kept at room temperature until the time of spinning (3000 rpm at 4 °C) and serum separation. Experiments were carried out with a cumulative dosing procedure, where doses of SalA (0.001–0.032 mg/kg, i.v.) or 16-BrSalA (0.0032–0.1 mg/kg, i.v.) were administered in increasing 0.5 log unit steps every 30 min, and a blood sample was taken 15 min after each dose. The KOR partial agonist nalorphine (0.1–3.2 mg/kg) was examined in an identical manner [97]. A repeated vehicle condition was studied under identical timing and sampling conditions. Prior studies have shown that such cumulative dose–effect curve procedures can be efficiently used in this assay to examine the potency and apparent efficacy of opioid ligands [97,101]. Serum samples were kept at −40 °C until the time of analysis, typically within 2 weeks of collection. Samples were analyzed in duplicate with a standard human prolactin immunoradiometric kit (MP Biomedicals; Solon, OH, USA), following the manufacturer’s instructions. Prolactin data were expressed as the change from the mean baseline (Δng/mL), via subtracting the mean value obtained at time −10 and −5 min, on each test day.

In an exploratory study, the sedation rating scores of 16-BrSalA and SalA were compared during the neuroendocrine experiments (just prior to blood sampling) via the use of a previously validated observational rating scale [59].

### 3.13. Rotating Disk Electrode Voltammetry

DA uptake in rat brain tissue suspensions was measured via the use of RDEV, as has been previously described [24,62]. Briefly, NAcc and dSTR tissues from drug-naïve rats were dissected, weighed, minced in an ice-cold KREBS buffer, and transferred to microcentrifuge vials. The tissues were then washed 8 times in a carbogen-aerated KREBS buffer at 37 °C, before being resuspended in 296 μL of KREBS and transferred to the RDEV chamber. A rotating glassy carbon electrode was lowered into the chamber and rotated at 2000 rpm with an MSR rotator (Pine instruments, AFMDO3GC, Durham, NC, USA). A +450 mV potential versus an Ag/AgCl reference electrode was applied, and the resulting current was measured with an eDAQ recorder (Denistone, NSW, Australia). Tissue suspensions were left to reach a stable baseline (≅10 min) before each test.

A low to infinite trans model was used to determine the uptake kinetics of the DAT, following sequential additions of increasing DA concentrations (0.5–4 μM, 4 μL) in tissue suspensions pretreated with 16-BrSalA (0 or 500 nM, 4 min pretreatment; *n* = 9/treatment/region). A zero trans model was used to determine DAT uptake following a single addition of DA (2 μM, 4 μL) in tissue suspensions pretreated with 16-BrSalA (0 or 500 nM, 4 min pretreatment) and nor-BNI (1 μM, 30 min pretreatment; *n* = 8–9/treatment/region). Uptake data were collected for 10 s, beginning 1 s after the addition of DA. A linear regression was calculated and normalized to a standard concentration curve in order to determine DA uptake, which was expressed as the pmol/s/g of tissue. For the low to infinite trans model, a Michaelis–Menten curve was fitted using GraphPad Prism (San Diego, CA, USA), and the V_max_ as well as K_m_ were determined for each suspension.

### 3.14. Imaging of ASP^+^ Uptake

The DAT uptake kinetics was also determined in HEK-293 cells coexpressing yellow fluorescent protein human DAT (YFP-hDAT) and rat myc-tagged-KOR (myc-rKOR) via measuring the uptake of the monoamine transporter substrate ASP^+^, as has been previously described [24,102,103]. Briefly, transfected cells were aspirated of a medium and washed twice in a KREBS buffer, before being placed in a stage-mounted incubator (37 °C) mounted within an Olympus Fluoview FV1000 confocal microscope (Sydney, NSW, Australia). Cells were preincubated with 16-BrSalA (0 or 10 μM) in 1 mL of a KREBS buffer for 2 min. The microscope was then focused on a cell monolayer, and the KREBS buffer was removed so that background autofluorescence could be determined via the capturing of a referencing image. ASP^+^ (1 mL, 1–16 μM) in a KREBS buffer containing 16-BrSalA (0 or 10 μM) was then added to different dishes. Cells were imaged every 5 s for 10 min to capture YFP (485 nm excitation, 545 nm emission) and ASP^+^ fluorescence (570 nm excitation, 670 nm emission). The linear slope of ASP^+^ accumulation was determined over a 60 sec period at the time of maximal effect, following correction for background fluorescence and normalization to YFP-hDAT expression (*n* = 17–51 cells/concentration/treatment). Data were entered into GraphPad Prism and Michaelis–Menten curves were fitted to determine the V_max_ as well as K_m_.

To determine the KOR- and ERK1/2-dependant manner of ASP^+^ uptake, separate dishes were preincubated with either nor-BNI (0 or 1 μM) or U0126 (0 or 20 μM) 30 min prior to the addition of a single concentration of ASP^+^ (1 mL, 10 μM) and, 5 min later, 16-BrSalA (0 or 10 μM). The linear slope of ASP^+^ accumulation was determined over a 60 sec period prior to the addition of 16-BrSalA and at the time of maximal effect. Cells without a linear uptake were discarded. The percentage change in the rate of ASP^+^ uptake was then calculated by comparing the change in the slope of ASP^+^ accumulation before and 16-BrSalA treatment (*n* = 38–65 cells/treatment).

### 3.15. ERK1/2 and p38 Western Blotting

For ex vivo experiments, drug-naïve rats were administered 16-BrSalA (1 mg/kg, i.p) either 0, 10, 15, 30, or 120 min prior to being sacrificed (*n* = 5–7/timepoint). The NAcc, dSTR, and PFC removed were then rapidly removed and homogenized in 100 μL (NAc) or 200 μL (dSTR and PFC) of an ice-cold RIPA buffer that contained both protease and phosphatase inhibitors. Tissue lysates were then centrifuged at 16,000× *g* for 15 min at 4 °C and the DNA pellet removed. For in vitro experiments, HEK-293 cells expressing myc-rKOR were serum-starved and incubated with 16-BrSalA (10 μM) for 0, 5, 10, 15, 30, 60, 120, or 180 min at 37 °C (*n* = 8 dishes/timepoint). To determine the KOR- and ERK1/2-mediated action of 16-BrSalA, another experiment was conducted where cells were pretreated with nor-BNI (0 or 1 μM), or U0126 (0 or 20 μM) for 30 min prior to a 10-minute incubation with 16-BrSalA (0 or 10 μM; *n* = 7–8 dishes/treatment). Following incubation, all of the cells were washed twice with ice-cold PBS, lysed with 200 μL of a RIPA buffer that contained protease and phosphatase inhibitors at 4 °C, and centrifuged at 16,000× *g* for 30 min at 4 °C.

The quantification of ERK1/2 and p38 was carried out as previously described [62]. Fifty micrograms of tissue or cell lysates was run on 10% SDS-PAGE gels. Following electrophoresis, separated proteins were transferred onto an Immobilon PVDF membrane, blocked with 5% bovine serum albumin (BSA) in tris-buffered saline (TBS) containing 0.1% Tween-20 (T-TBS) for 1 h at room temperature, and then probed with the primary antibody diluted in a blocking buffer overnight (1:500 mouse monoclonal p-ERK1/2, 1:1000 rabbit monoclonal ERK1/2, 1:1000 rabbit monoclonal p-p38, and 1:1000 rabbit monoclonal p38). The membrane was then washed 3 times with T-TBS before being probed with the secondary antibody (1:5000 goat anti-mouse Cy5, 1:5000 goat anti-rabbit Cy5) diluted in TBS at room temperature for 1 h. The membrane was then washed 3 times with TBS and scanned via the use of a FUJIFILM FLA-5000 scanner (Fujifilm, Tokyo, Japan). Membranes were first probed and scanned for the phosphorylated protein, before being stripped, probed, and scanned for the total protein (stripping buffer: 0.05 M Tris-HCl, 2% SDS, and 3.75% β-mercaptoethanol, pH 6.8, incubated at 40 min at room temperature followed by five–seven washes in T-TBS).

All Western blots were analyzed via the use of ImageJ (NIH). All band densities were background-corrected and corrected for protein loading differences through normalizing the density of phosphorylated protein bands to the corresponding total protein band. The data were normalized to vehicle-treated controls to enable comparisons between membranes. The size of visual protein bands was quantified via the use of their linear migration properties to enable the identification of proteins. Phosphorylated ERK1/2 and p38 expression was normalized to total ERK1/2 and p38, respectively, and expressed as fold change from the baseline (time 0/vehicle + vehicle treatment).

### 3.16. Statistical Analysis

Statistical analyses were carried out via the use of GraphPad Prism (v9.1.0, La Jolla, CA, USA) with the recommended parameters. Experiments with time course data were analyzed via the use of a two-way (treatment × time) analysis of variance (ANOVA), with time as a repeated measure. Sphericity was not assumed, and thus Greenhouse–Geisser corrections to degrees of freedom were applied. Experiments without time course data were analyzed via the use of one-way ANOVAs (repeated measures where appropriate) or *t*-tests. Significant interactions or effects of treatment were followed-up with multiple-comparison testing using the recommended tests., i.e., Šídák’s test for comparing treatments vs. controls, Dunnett’s test for comparing multiple treatment groups vs. controls, or Tukey tests for comparing all of the groups in experiments with antagonists/inhibitors. Normalized Western blot data were analyzed via the use of one-sample *t*-tests. The results were considered significant when *p* < 0.05.

## 4. Conclusions

The KOR has been identified as a promising target for the management of psychostimulant use disorders and for the development of non-addictive pain medications. Unfortunately, KOR agonists are typically associated with various adverse side effects that limit their clinical viability. In the current study, we showed that the SalA analogue 16-BrSalA had significant anticocaine effects in drug-seeking and locomotor hyperactivity models. We showed that 16-BrSalA produced an ERK-dependent increase in DAT activity, which likely underlies these anticocaine effects. Importantly, 16-BrSalA produced minimal side effects, with no significant effects on all tests other than conditioned aversion. Furthermore, we showed that 16-BrSalA increased early phase ERK1/2 phosphorylation and had a small effect on p38 phosphorylation. These results provide support for the idea that KOR agonists with differential signaling can be developed to dissociate desirable therapeutic effects from side effects. More research into the development of non-aversive KOR agonists is still needed, however.

## Figures and Tables

**Figure 1 molecules-28-04848-f001:**
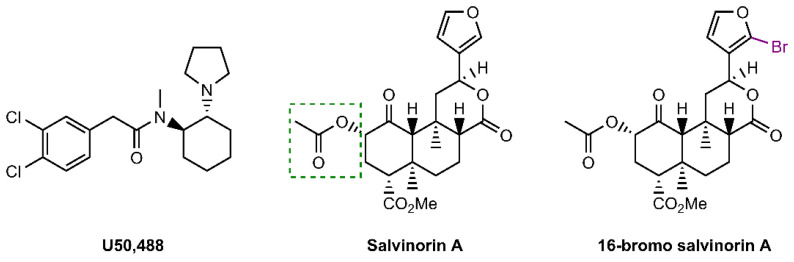
Chemical structures of U50,488, salvinorin A, and 16-bromo salvinorin A. The green dashed box indicates the C2 position of previously investigated analogues of salvinorin A [65].

**Figure 2 molecules-28-04848-f002:**
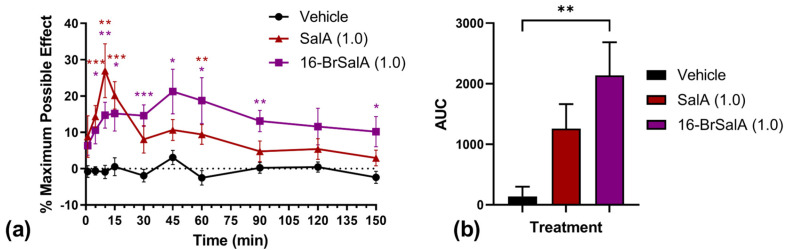
Results of the warm water tail withdrawal assay in vehicle-, salvinorin A (SalA; 1.0 mg/kg)-, or 16-bromo SalA (16-BrSalA; 1.0 mg/kg)-treated mice (*n* = 8–11/treatment). (**a**) Effect of treatment on % maximal possible antinociceptive effect as a function of time. (**b**) Area under the curve (AUC) data for each treatment. * *p* < 0.05, ** *p* < 0.01, and *** *p* < 0.001 compared to the vehicle (Dunnett’s test).

**Figure 3 molecules-28-04848-f003:**
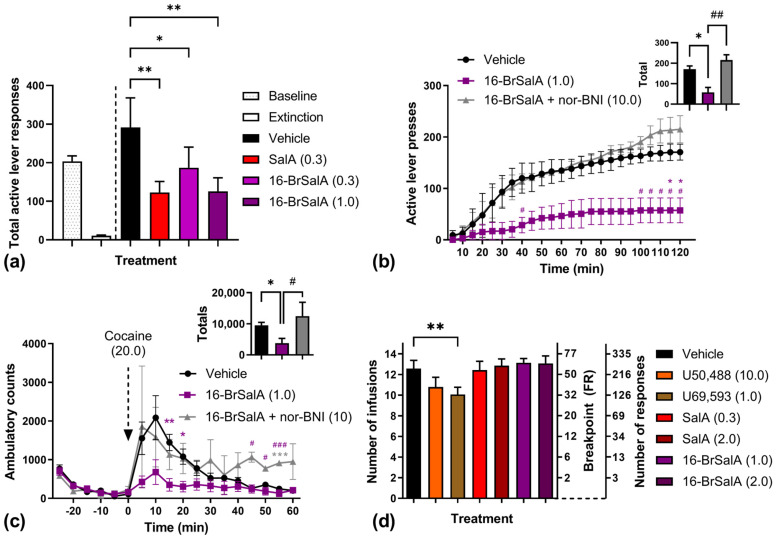
Anticocaine effects of 16-BrSalA. (**a**) Total active lever responding during cocaine (20 mg/kg) + cue-induced reinstatement following vehicle, SalA (0.3 mg/kg), or 16-BrSalA (0.3, 1.0 mg/kg) treatment (*n* = 6; data partially reproduced from [60]). (**b**) Time course of cocaine (20 mg/kg) + cue reinstatement data (with an inset showing summed totals) displaying the effect of nor-binaltorphimine (nor-BNI; 10 mg/kg) on 16-BrSalA (1 mg/kg) treatment (*n* = 3). (**c**) Cocaine-induced (20 mg/kg) ambulatory counts (with an inset showing summed totals) in vehicle- and 16-BrSalA (1.0 mg/kg)-treated rats (*n* = 6/treatment). (**d**) Total number of cocaine (1 mg/kg) infusions (and break point/number of responses) earned on a progressive ratio schedule of reinforcement following vehicle, U50,488 (10.0 mg/kg), U69,593 (1.0 mg/kg), SalA (0.3, 1.0 mg/kg), or 16-BrSalA (1.0, 2.0 mg/kg) treatment (*n* = 14). * *p* < 0.05, ** *p* < 0.01, and *** *p* < 0.001 compared to the vehicle (Dunnett [**a**,**d**] and Tukey [**b**,**c**] tests). ^#^
*p* < 0.05, ^##^
*p* < 0.01, and ^###^
*p* < 0.001 compared to + nor-BNI (Tukey test).

**Figure 4 molecules-28-04848-f004:**
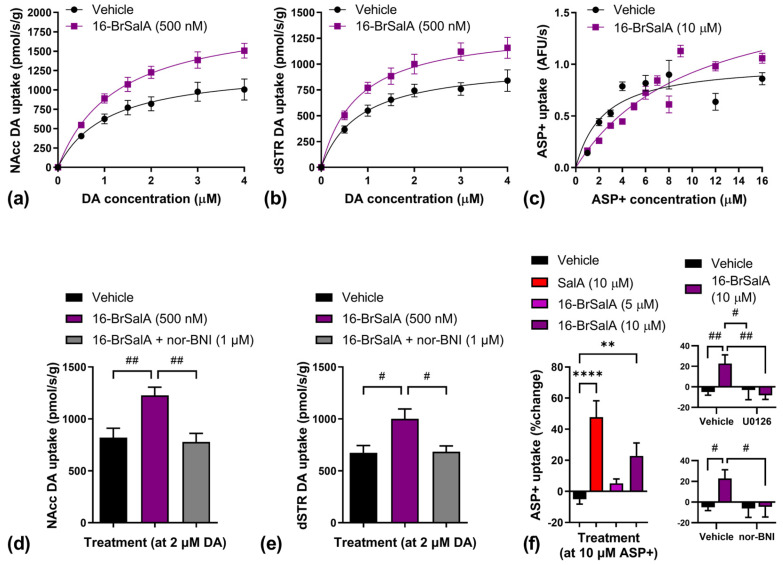
Effect of 16-BrSalA on dopamine (DA) transporter (DAT) function. DA uptake kinetics over 0–4 μM concentrations in vehicle- and 16-BrSalA (500 nM)-treated nucleus accumbens (NAcc) (**a**) as well as dorsal striatal (dSTR) (**b**) tissue suspensions as measured by rotating disk electrode voltammetry (RDEV) using a low to infinite trans model (*n* = 9 samples/region/treatment). (**c**) 4-[4-(dimethylamino)styryl]-*N*-methylpyridinium (ASP^+^) uptake kinetics over 0–16 μM concentrations in vehicle- and 16-BrSalA (10 μM)-treated HEK-293 cells coexpressing the DAT and kappa opioid receptor (KOR) as measured via time-resolved fluorescence imaging (auto-fluorescence units/s (AFU/s), *n* = 17–51 cells/concentration/treatment). Effect of nor-BNI (1 μM) treatment on 16-BrSalA (500 nM)-induced change in uptake of 2 μM DA in rat NAcc (**d**) and dSTR (**e**) tissue suspensions as measured through RDEV using the zero trans model (*n* = 8–9 sections/region/treatment). (**f**) SalA (10 μM)- and 16-BrSalA (5, 10 μM)-induced change in uptake of 10 μM ASP^+^ in HEK-293 cells coexpressing the DAT and KOR as well as the effect of pretreatment with nor-BNI (1 μM) or U0126 (20 μM) (*n* = 38–65 cells/treatment). ** *p* < 0.01, **** *p* < 0.0001 (Dunnett’s test). ^#^
*p* < 0.05, ^##^
*p* < 0.01 (Tukey test).

**Figure 5 molecules-28-04848-f005:**
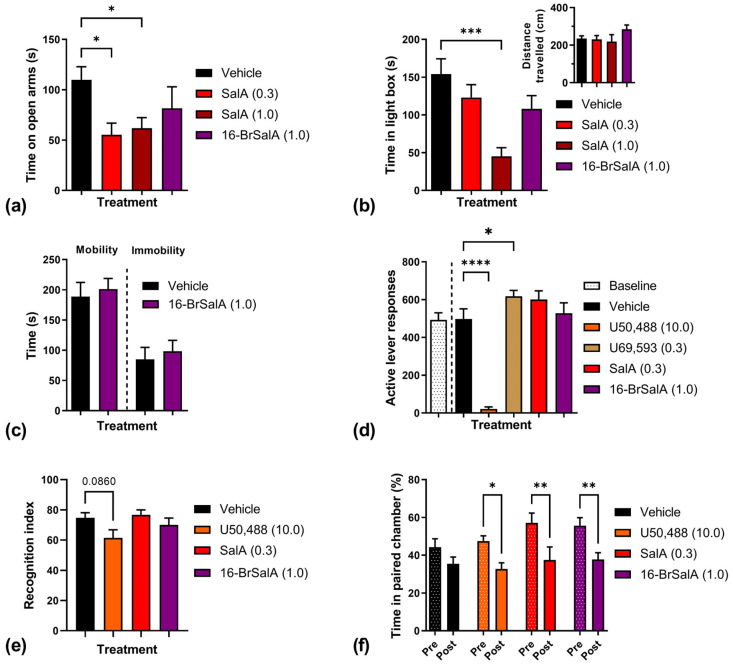
Effect of 16-BrSalA compared to SalA and other KOR agonists in various preclinical side effect assays. (**a**) Time spent on the open arms of the elevated plus maze as a function of vehicle, SalA (0.3, 1.0 mg/kg), or 16-BrSalA (1.0 mg/kg) treatment (*n* = 15–29/treatment). (**b**) Time spent in the light box in the light–dark test (with an inset showing the total distance travelled) as a function of vehicle, SalA (0.3, 1.0 mg/kg), or 16-BrSalA (1.0 mg/kg) treatment (*n* = 9–24/treatment). (**c**) Mobility and immobility time in the forced swim test as a function of vehicle or 16-BrSalA (1.0 mg/kg) treatment (8–11/treatment). (**d**) Total active lever responses maintained by sucrose reinforcement following vehicle, U50,488 (10.0 mg/kg), U69,593 (0.3 mg/kg), SalA (0.3 mg/kg), or 16-BrSalA (1.0 mg/kg) treatment (*n* = 7). (**e**) Recognition index in the novel object recognition test as a function of vehicle, U50,488 (10.0 mg/kg), SalA (0.3, 1.0 mg/kg), or 16-BrSalA (1.0 mg/kg) treatment (*n* = 21). (**f**) Percentage of time spent in the paired chamber pre- and postconditioning with the vehicle, SalA (0.3), or 16-BrSalA (1.0 mg/kg) (*n* = 8–11/treatment). * *p* < 0.05, ** *p* < 0.01, *** *p* < 0.001, and **** *p* < 0.0001 (Dunnett’s [**a**–**e**] and Šídák’s [**f**] tests). Note that the data from the control groups have been published previously [65].

**Figure 6 molecules-28-04848-f006:**
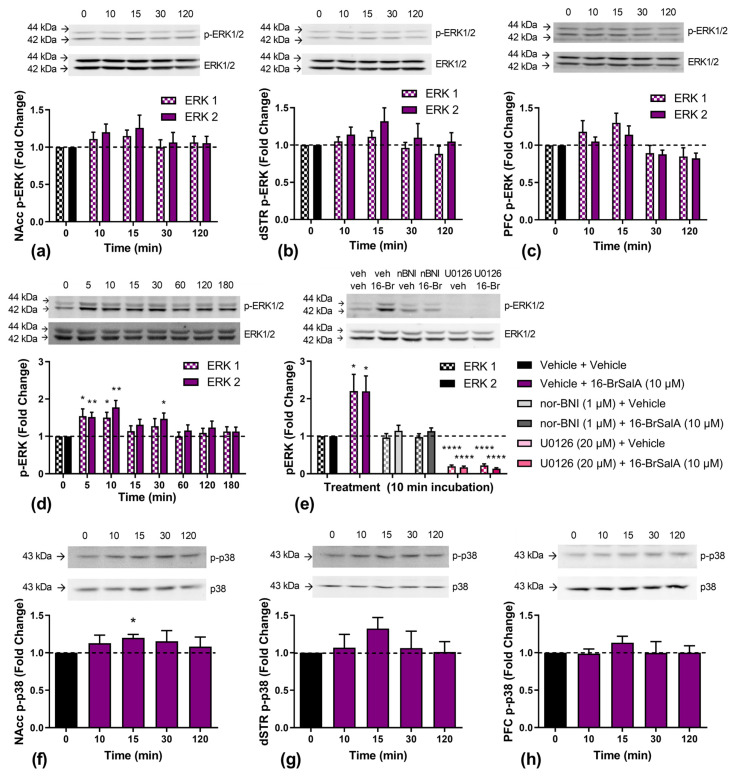
Effect of 16-BrSalA on extracellular-signal-regulated kinases 1 and 2 (ERK1/2) and p38. Phosphorylated ERK1/2 (p-ERK1/2) was quantified in rat NAcc (**a**), dSTR (**b**), and prefrontal cortex (PFC) (**c**) tissue collected 0–120 min following treatment with 16-BrSalA (*n* = 5–7/timepoint/region). (**d**) p-ERK1/2 was also measured in HEK-293 cells coexpressing the DAT and KOR 0–180 min following treatment with 16-BrSalA (*n* = 8 dishes/timepoint). (**e**) Effect of treatment with nor-BNI (1 μM) or U0126 (20 μM) on 16-brSalA (10 μM)-induced ERK1/2 expression in HEK-293 cells coexpressing DAT and KOR (*n* = 7–8 dishes/treatment). Phosphorylated p38 (p-p38) was similarly measured in rat NAcc (**f**), dSTR (**g**), and PFC (**h**) tissue collected 0–120 min following treatment with 16-BrSalA (*n* = 5–7/timepoint/region). Representative Western blot scans are displayed above each graph. p-ERK1/2 and p-p38 expression were normalized to total ERK1/2 and p38, respectively, and expressed as fold change from the baseline (time 0/vehicle + vehicle treatment). * *p* < 0.05, ** *p* < 0.01, and **** *p* < 0.0001 compared to the baseline (one-sample *t*-test).

**Figure 7 molecules-28-04848-f007:**
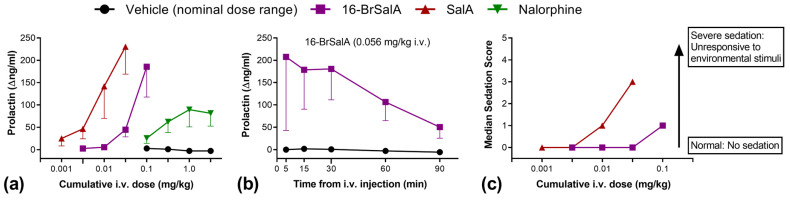
Effect of 16-BrSalA, SalA, and nalorphine in rhesus monkeys (*n* = 3). (**a**) Cumulative dose effect of 16-BrSalA, SalA, nalorphine, and vehicle on mean (±SEM) serum levels of prolactin. Prolactin levels are expressed as change from pre-injection values (Δng/mL). (**b**) Time course of mean (±SEM) blood prolactin levels following 16-BrSalA administration (0.056 mg/kg, i.v.). (**c**) Cumulative dose effect of 16-BrSalA and SalA on the median sedation score.

**Table 1 molecules-28-04848-t001:** Effect of 16-BrSalA on the kinetics of DA uptake (pmol/s/g) in rat NAcc/dSTR tissue as well as ASP^+^ uptake (AFU/s) in HEK-293 cells coexpressing DAT and KOR.

Region	Treatment	V_max_	K_m_
NAcc	Vehicle	1356.16 ± 215.36	1.17 ± 0.28
	16-BrSalA	1965.33 ± 131.65 *	1.28 ± 0.15
dSTR	Vehicle	1034.36 ± 126.44	0.90 ± 0.18
	16-BrSalA	1433.44 ± 24.00 *	0.94 ± 0.14
In vitro	Vehicle	1.02 ± 0.08	2.41 ± 0.51
	16-BrSalA	1.84 ± 0.19 ****	10.05 ± 1.89 ****

Values are mean ± SEM. * *p* < 0.05, **** *p* < 0.0001 compared to vehicle treatment (*t*-test).

## Data Availability

Data are presented within the manuscript. Additional raw data are available on request from the corresponding author.

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
