# Peer review of "The Kappa Opioid Receptor Agonist 16-Bromo Salvinorin A Has Anti-Cocaine Effects without Significant Effects on Locomotion, Food Reward, Learning and Memory, or Anxiety and Depressive-like Behaviors"

_molecules, 2023, doi:10.3390/molecules28124848_

Round 1

Reviewer 1 Report

The authors evaluated the anti-cocaine effect of 16-BrSalA and its potential side effects. The results suggested that 16-BrSalA had more potent anti-cocaine effects with longer duration and with fewer side effects compare to the parent compound SalA. These seem useful information. However, the authors should address the following minor points before publication.

            “Confirmation (line 65)” is not correct. It would be a typo of “conformation.

            The authors described “modification at the C-2 position (line 82).” I recommend that the C-2 position is indicated in the SalA structure in figure 1.

            The abbreviation “PR” firstly appeared in line 130. So, its spelling should be shown.

            The authors described “both compounds having similar efficacy (line 139).” Indeed, the refence 60 said in the main text that most derivatives retained full efficacy for KOR activity. However, Table 1 in the reference 60 showed EC50 values. Is “potency” correct? Moreover, I recommend that numerical concrete values are shown.

Author Response

We thank the reviewer for the suggested modifications to our manuscript that has improved the quality.

  1. “Confirmation (line 65)” is not correct. It would be a typo of “conformation.
    • Corrected
  1. The authors described “modification at the C-2 position (line 82).” I recommend that the C-2 position is indicated in the SalA structure in figure 1.
    • The figure and figure caption has been updated to indicate this.

  1. The abbreviation “PR” firstly appeared in line 130. So, its spelling should be shown.
    • Corrected

  1. The authors described “both compounds having similar efficacy (line 139).” Indeed, the refence 60 said in the main text that most derivatives retained full efficacy for KOR activity. However, Table 1 in the reference 60 showed EC50 Is “potency” correct? Moreover, I recommend that numerical concrete values are shown. 
    • We have now  given detailed  affinity, efficacy, and potency data for SalA and 16-BrSalA from references 60 and 67 (new), in the last paragraph of the introduction.

Reviewer 2 Report

Review: molecules-2408394: Kappa opioid receptor (KOR) agonists have potential value in treating psychostimulant abuse, but agonists like Salvinorin A (SalA) show rapid metabolism, aversion and other liabilities of use that limit their therapeutic value. Following up their earlier studies (Paton et al., Front Neurosci., 2020 and Front Pharmacol., 2022), van de Wetering and colleagues here test 16-bromo salvinorin A (16-BrSalA) for duration of antinociceptive activity in a warm-water tail withdrawal test, and dose-dependently decreases cocaine-primed reinstatement of cocaine self-administration in a nor-BNI-sensitive manner.  Curiously, neither SalA or 16BrSalA significantly effected responding for cocaine in a progressive ratio schedule, but the addition of mechanistic studies of 16-BrSalA impact on dopamine levels and DAT activity in brain homogenates and the discounting of potential alternatives arising from potential adverse liabilities (anxiety, depression and learning and memory performance, as well as conditioned place aversion). Incorporation of the KOR antagonist nor-BNI and comparison to both SalA and other KOR agonists across a terrific battery of tests across assays and species increases the impact and significance of this work.  The results are supported by appropriate statistical analysis and the conclusions reached generally supported by the data (although a few concerns remain; see below).  Notably, the authors also demonstrate 16-BrSalA stimulates prolactin release in monkeys, a confirmation of KOR agonism in primates that strengthens the relevance of the study.  The authors take some pains to demonstrate in cellular assays 16-BrSalA biased agonism, with minimal ERK and p38 activation only in the early phase consistent with reports of g-protein biased signaling.  Although modestly off the focus on cocaine therapeutics, these studies do lend themselves to explain the aversive effects reported, and as such additionally support the overall conclusions that 16-BrSalA has therapeutic potential as a treatment for cocaine abuse.  While the work presented sometimes repeats earlier findings to reduce impact, it is well presented and the conclusions fully supported by the presented data, resulting in a strong manuscript and enthusiasm for an interesting and timely study. 

Concerns (and to be clear, these are deemed modest only):

16-BrSalA has been characterized previously as producing antinociception longer than the parent SalA compound (Paton et al., Front Neurosci., 2020), modestly reducing impact of the increased duration in the present study.  Although the authors do appropriately acknowledge the earlier results, the results are somewhat overstated with the repetition and comparison to the presently untested 16-ethynylSalA. The manuscript might be strengthened by softening this element, or adding the pharmacokinetics data necessary to support claims of the importance of modification at the C16 position of SalA for a prolonged PK profile.

16-BrSalA has been characterized previously as a KOR biased agonist favoring G-protein activation (Paton et al., Front Neurosci., 2020).  Although the authors confirm that activity here in cellular assays, inspection of the data suggests 16-BrSalA may simply act as KOR partial agonist (further influenced by Fig. 3 in Paton et al., 2020).  The absence of a full dose-response test, with appropriate measures of comparison between drug administrations (i.e., EC50 and ED50 data, with matching 95% confidence intervals) and appropriate statistical analysis (nonlinear regression analysis) weakens this otherwise important demonstration.  Without it, the alternative explanation for all results (that an insufficient dose was tested) cannot be discounted. Addition of (or literature review of) a full dose-response for this compound and the evidence of therapeutic benefit from KOR partial agonists would strengthen the manuscript.

Notably, 16-BrSalA was shown to produce significant locomotor impairment (Paton et al., Front Neurosci., 2020).  Commendably, the authors work around these limitations in the present study, using lower doses to avoid locomotor impairment, and additionally confirm 16-BrSalA-mediated reversal of cocaine-impaired dopamine levels and DAT activity to confirm the anti-cocaine effects of 16-BrSalA, turning a potential weakness into a strength of this manuscript.

Author Response

We would like to thank the reviewer for their suggested improvements to our manuscript. 

  1. 16-BrSalA has been characterized previously as producing antinociception longer than the parent SalA compound (Paton et al., Front Neurosci., 2020), modestly reducing impact of the increased duration in the present study.  Although the authors do appropriately acknowledge the earlier results, the results are somewhat overstated with the repetition and comparison to the presently untested 16-ethynylSalA. The manuscript might be strengthened by softening this element, or adding the pharmacokinetics data necessary to support claims of the importance of modification at the C16 position of SalA for a prolonged PK profile. 
    • We have edited the abstract and final conclusion to avoid repeating this finding in these key sections so as to soften this aspect of the study, as suggested.

  1. 16-BrSalA has been characterized previously as a KOR biased agonist favoring G-protein activation (Paton et al., Front Neurosci., 2020).  Although the authors confirm that activity here in cellular assays, inspection of the data suggests 16-BrSalA may simply act as KOR partial agonist (further influenced by Fig. 3 in Paton et al., 2020).  The absence of a full dose-response test, with appropriate measures of comparison between drug administrations (i.e., EC50 and ED50 data, with matching 95% confidence intervals) and appropriate statistical analysis (nonlinear regression analysis) weakens this otherwise important demonstration.  Without it, the alternative explanation for all results (that an insufficient dose was tested) cannot be discounted. Addition of (or literature review of) a full dose-response for this compound and the evidence of therapeutic benefit from KOR partial agonists would strengthen the manuscript.
    • 16-BrSalA is a full agonist of the KOR in vitro. We have now more thoroughly reviewed the in vitro dose effect data (affinity, potency, efficacy) for 16-BrSalA and SalA in the last paragraph of the introduction (page 2, lines 86-94).
    • The reason for the lower efficacy of 16-BrSal in the antinociception assay (Fig 3, Paton et al 2020) has not been fully explored but could be due to several factors such as drug metabolism or signalling bias.
    • Carrying out a full dose effect assay for drug seeking tests is not feasible, but we do believe the doses used in our study were appropriate. The dose of 16-BrSalA (1.0 mg/kg) evaluated in the side effect assays was the dose required to produce significant anti-cocaine effects that were also equivalent to SalA (0.3 mg/kg). Therefore, the side effect assays evaluated if 16-BrSalA produced less side effects than SalA at noequivalent ‘therapeutic’ doses. We have now made this clearer in the manuscript (first paragraph of Section 2.4).

  1. Notably, 16-BrSalA was shown to produce significant locomotor impairment (Paton et al., Front Neurosci., 2020).  Commendably, the authors work around these limitations in the present study, using lower doses to avoid locomotor impairment, and additionally confirm 16-BrSalA-mediated reversal of cocaine-impaired dopamine levels and DAT activity to confirm the anti-cocaine effects of 16-BrSalA, turning a potential weakness into a strength of this manuscript.
    • No comment to address

Reviewer 3 Report

This manuscript sufficiently includes pharmacological data about 16-bromo salvinorin A (16-BrSalA) to demonstrate the anti-cocaine effects.  This reviewer thought the authors properly explained them in detail. Judging from the experimental results, 16-BrSalA is probably a more useful compound than salvinorin A (SalA) for the development of new analgesics with kappa opioid receptor (KOR) agonistic activities.  Especially, the authors’ data of Figure 4 are significant in this manuscript because they indicate that 16-BrSalA has a potential to suppress its side effects which are induced by dopamine (DA).  In addition, the authors also showed the effects of 16-BrSalA on ERK1/2 and p38 in Figure 6 to investigate the cell signaling triggered by KOR agonists.  Therefore, the reviewer recommends this manuscript as an acceptable ‘review article’ in “Molecules”, after the authors respond to the following points.  

 [Major]

1)     The authors need to add comments about active conformations of 16-BrSalA.  Especially, they are necessary to deduce the orientations of the furan rings in both 16-BrSalA and SalA. What it the difference between them? 

2)     How does the bromination onto the furan ring of SalA affect the pharmacological activities of 16-BrSalA?  

3)     In Figure 5, the results of 16-BrSalA did not significantly show the deference from those of SalA. This reviewer simply wonders the results are useful in this manuscript. 

4)     In the manuscript title, ‘minimal side effects’ is probably short on specifics. This reviewer recommends the authors to change it to more suitable phrases.

 [Minor]

1)     In the title of Figure 1, the authors should revise ‘structure’ to ‘structures.  

2)     How different is the activation (phosphorylation) of ERK1 from that of ERK-2 in the KOR-mediated cellular signaling? 

3)     In Figure 6, several letters are too small to read clearly. The authors must revise them.

I do not have any comments about the quality of English in the manuscript without my comments as described above.

Author Response

We thank the reviewer for suggesting the improvements to our manuscript. We have addresses these as follows:

[Major]

  1. The authors need to add comments about active conformations of 16-BrSalA.  Especially, they are necessary to deduce the orientations of the furan rings in both 16-BrSalA and SalA. What it the difference between them? 
    • Orientation of the furan ring was shown and explored previously in ref 60, 66(new), and 67(new), which we have now referenced when introducing 16-BrSal A in the last paragraph of the introduction.

  1. How does the bromination onto the furan ring of SalA affect the pharmacological activities of 16-BrSalA?  
    • We have now more thoroughly reviewed the pharmacological activities for 16-BrSalA and SalA in the last paragraph of the introduction.

  1. In Figure 5, the results of 16-BrSalA did not significantly show the deference from those of SalA. This reviewer simply wonders the results are useful in this manuscript. 
    • We believe these findings are still meaningful since they show that a dose of 16-brSalA that is capable of significantly reducing cocaine-induced drug seeking does not produce significant side effects (other than aversion).

  1. In the manuscript title, ‘minimal side effects’ is probably short on specifics. This reviewer recommends the authors to change it to more suitable phrases.
    • We thank the reviewer for this suggestion, however, if we include full details of side effects the title will be too long.

[Minor]

  1. In the title of Figure 1, the authors should revise ‘structure’ to ‘structures.  
    • Corrected
  1. How different is the activation (phosphorylation) of ERK1 from that of ERK-2 in the KOR-mediated cellular signaling? 
    • ERK1 and 2 have not been found to have any specific roles(https://doi.org/10.3389/fcell.2016.00053) and do not impact KOR-mediated signalling differentially. We have added this detail to lines 344-345.
  1. In Figure 6, several letters are too small to read clearly. The authors must revise them.
    • We have increased the size of the smaller fonts in this figure (size 8 minimum).